# Using Human-Centered Design in Community-Based Public Health Research: Insights from the ECHO Study on COVID-19 Vaccine Hesitancy in Montreal, Canada

**DOI:** 10.3390/ijerph22020198

**Published:** 2025-01-30

**Authors:** Krystelle Marie Abalovi, Geneviève Fortin, Maryam Parvez, Joyeuse Senga, Joe Abou-Malhab, Cat Tuong-Nguyen, Caroline Quach, Ashley Vandermorris, Kate Zinzser, Britt McKinnon

**Affiliations:** 1École de Santé Publique, Université de Montréal, Montréal, QC H3N 1X9, Canada; krystelle.marie.abalovi@gmail.com (K.M.A.); genevieve.fortin.6@umontreal.ca (G.F.); cat.tuong-nguyen@msss.gouv.qc.ca (C.T.-N.); 2Epidemiology, Biostatistics and Occupational Health, McGill University, Montreal, QC H3A 1Y7, Canada; maryam.parvez@mail.mcgill.ca (M.P.); joyeuse.senga@mail.mcgill.ca (J.S.); aboumalhabjoe@gmail.com (J.A.-M.); 3Departments of Microbiology, Infectious Diseases and Immunology & Pediatrics, University of Montreal, Montreal, QC H3T 1J4, Canada; c.quach@umontreal.ca; 4Division of Adolescent Medicine, Department of Pediatrics, The Hospital for Sick Children, Toronto, ON M5G 1X8, Canada; ashley.vandermorris@sickkids.ca; 5Centre de Recherche en Santé Publique, Montréal, QC H3N 1X9, Canada; britt.mckinnon@utoronto.ca

**Keywords:** human-centered design, HCD, vaccine hesitancy, COVID-19, community-based research, public health

## Abstract

(1) Background: This study used human-centered design (HCD) within a community-based research project to collaboratively develop local strategies aimed at enhancing COVID-19 vaccine confidence among children and youth. (2) Methods: HCD projects were carried out between December 2021 and August 2022 by four community-based design (CBD) teams in Montreal, Canada. The CBD teams were composed of parent and youth community members, public health and social science researchers, and HCD specialists. Process evaluation data, collected from the CBD team members through focus group discussions and written questionnaires, were used to reflect on the use of HCD in this project. (3) Results: The CBD teams designed and implemented projects addressing factors they identified as contributing to COVID-19 vaccine hesitancy for children and youth in their communities, including misinformation, lack of trust, social inequities, and resistance to pandemic-related restrictions. The CBD team members appreciated many aspects of the HCD approach, especially the values it stands for, such as empathy, co-creation, and collaboration. HCD and public health specialists described some tension between the different disciplinary approaches. (4) Conclusions: HCD holds promise for addressing complex public health issues, though further exploration of strategies for integrating HCD within established models of community-based public health research is needed.

## 1. Introduction

Human-centered design (HCD) is a creative problem-solving approach that involves building deep empathy with the people for whom a design is intended, generating numerous ideas, building prototypes, and sharing these prototypes with the target population to refine and implement innovative solutions [1]. Traditionally more popular in the private sector, HCD has been gaining traction as an approach for solving public health issues, particularly among underserved populations [2,3]. HCD has been applied in various areas of public health, including breastfeeding programs, nutrition education videos, and digital health technologies [4,5]. Specifically, it has been successfully utilized to address vaccine hesitancy by developing tailored vaccine promotion materials, communication strategies, and educational tools designed to counteract misinformation and resistance to health guidelines [6,7,8].

While prior research has demonstrated HCD’s value in creating individual-level interventions, such as digital tools and clinician-focused strategies, its use in addressing systemic and structural barriers to vaccine hesitancy remains underexplored [6,7,8]. Most applications have focused on improving usability and decision-making for specific stakeholders, but few studies have examined how HCD can integrate community-driven solutions to tackle the broader social determinants influencing vaccine hesitancy, such as distrust in the healthcare system, misinformation, and social inequities. These gaps highlight the need for more evidence on the feasibility and effectiveness of HCD in addressing vaccine hesitancy as a complex public health issue.

Research suggests that HCD can significantly improve public health outcomes by creating innovative, culturally tailored interventions that meet the specific needs of communities [9,10,11]. However, implementing HCD in public health also faces several challenges, including resource constraints, the need for effective interdisciplinary collaboration, and difficulties in scaling prototypes [10,12]. The success of HCD projects also depends on comprehensive and representative community participation; without it, the proposed solutions may fail to effectively address broader community needs. Addressing these challenges is essential for maximizing HCD’s potential in public health research and interventions.

Vaccine hesitancy, defined as the reluctance or refusal to be vaccinated despite the availability of vaccines, represents a significant global health challenge that has been further exacerbated by the COVID-19 pandemic [13]. Decisions about vaccination are influenced by trust-related historical, sociocultural, and political factors, often leading to disparities in vaccine hesitancy among different populations [14]. In high-income countries, vaccine hesitancy is often more prevalent among young people, those with lower socioeconomic status, immigrants, and racialized communities [15]. These disparities are rooted in systemic barriers and differential access to health resources and can reflect the broader impact of social inequities on health decisions [16]. Compounding these issues, misinformation and disinformation have played a critical role in shaping negative attitudes toward vaccination, undermining trust in healthcare systems and perpetuating false narratives about vaccine safety and efficacy [17]. Addressing these challenges requires a multifaceted approach that includes combating misinformation and disinformation, as well as fostering community engagement, which has been shown to be effective in establishing trust, ensuring cultural relevance, and addressing distinct community-specific concerns [18].

In response to these challenges, the ECHO community study on COVID-19 vaccine hesitancy (Étude Communautaire sur l’Hésitation à la vaccination contre la COVID-19) was initiated. The ECHO study had two main objectives: (1) to monitor COVID-19 vaccination intention and uptake among adolescents and parents of primary school children using school-based surveys and (2) to use HCD to develop and pilot community-driven solutions to enhance vaccine confidence among parents and youth. This paper focuses on the second objective (the results for objective 1 are reported elsewhere [19]). We describe the results of four HCD projects and, through a process evaluation, critically reflect on the projects’ use of HCD from the perspectives of community researchers, HCD specialists, and public health students and professionals. Through these efforts, we aim to illustrate the potential of HCD in addressing vaccine hesitancy and provide insights into the challenges and successes encountered in the process.

## 2. Methods

### 2.1. Context and Setting

A detailed description of the study methodology is available in the published protocol [20]. Briefly, this study was conducted in two ethnoculturally diverse, lower-income neighborhoods in Montreal, the largest city in the Canadian province of Quebec. These neighborhoods were selected because they were disproportionately affected by the pandemic’s economic and health-related consequences [21,22] and had a lower initial uptake of COVID-19 vaccination [23,24].

The HCD process began in December 2021, coinciding with the initial Omicron wave and the initiation of COVID-19 vaccination for children aged 5–11 years. At that time, youth aged 12–17 years had been eligible for vaccination since May 2021 [25]. Quebec’s vaccine passport program mandated that individuals aged 13 and above show proof of vaccination to enter nonessential venues such as restaurants, cinemas, and gyms as well as participate in various extracurricular activities and sports [15]. Evidence suggests the provincial vaccine passport policy slightly increased adolescent vaccination rates by 2.3 percentage points; [26] however, many adolescents also expressed strong resistance to the policy [19]. In Quebec, individuals can make autonomous health decisions starting at 14 years of age, meaning that adolescents aged 14 and above did not legally require their parents’ or guardians’ permission to receive the vaccine [27].

### 2.2. HCD Process

Four community-based design (CBD) teams, composed of four members each, participated in the HCD process. There was one youth-led and one parent-led team from each neighborhood. Adolescents aged 14–17 years and parents were recruited for CBD teams through local community organizations, schools, and social media. The adolescent and parent teams worked in parallel on independent HCD projects, each addressing distinct challenges and target populations. However, the teams participated together in training workshops, where they shared insights and strategies to enhance their respective projects.

CBD team members were compensated for a 5 h weekly commitment to the project. Each team was mentored by a public health graduate student, with four mentors in total, one assigned to each team. These mentors, along with the project coordinator, two HCD specialists, and researchers from a variety of disciplines (e.g., epidemiology, anthropology, adolescent medicine, and pediatric infectious diseases), formed the “support team”. This group provided guidance, expertise, and resources throughout the HCD process, ensuring that each CBD team was well-supported.

The HCD process had three phases: inspiration, ideation, and implementation. The process took six months, but some teams continued to implement their interventions for up to an additional three months. A summary of the objectives and key activities for each phase are shown in Table 1; additional details of the phases have been described elsewhere [20].

Briefly, in the inspiration phase (December 2021 to March 2022), CBD teams began by gathering information on COVID-19 vaccination decision-making in their communities through existing data and local resources. The teams then conducted semi-structured interviews with parents of unvaccinated children aged 5–11 years and unvaccinated youth (potential “users” in the HCD process), using purposive and convenience sampling methods. Adolescent teams conducted 25 interviews, all of which were conducted in French. Parent team conducted 16 interviews in French, English, or Arabic, which were translated into French or English by the teams. Interviews lasted an average of 20–30 min, and guides are available in the Appendix A. Training in interviewing techniques was provided, and interviews were conducted in pairs (with one interviewer and one note-taker), either in person or via Zoom, ensuring privacy and confidentiality.

The teams synthesized interview data using empathy maps—visual representations of the attitudes, thoughts, and feelings of their target group—to create personas that helped guide the design process [20]. Personas are fictional characters created to represent the needs and motivations of a group of users with shared characteristics. They humanize the design process and capture the essence of interview data, helping to frame a design challenge (Figure 1 and Figure 2). In the ideation phase (March–April 2022), teams generated potential solutions through brainstorming and developed prototypes, which were then iteratively tested with users. Finally, in the implementation phase (May–June 2022), prototypes were finalized, created, and piloted among the target populations, with plans for monitoring and evaluation using both quantitative and qualitative indicators. Each team was provided a budget of CAD $15,000 for the implementation phase.

Throughout the HCD process, CBD teams met virtually with mentors at least weekly and participated in monthly in-person and virtual training workshops, facilitated by HCD experts and supported by researchers and other individuals with local and substantive expertise. A total of six 5 h training workshops were held: four in person and two virtual due to public health restrictions. Topics covered in each session are shown in Table 1.

### 2.3. Process Evaluation

To better understand the challenges and opportunities with respect to using HCD in the ECHO project, we documented and critically reflected on the process by analyzing data collected from CBD team members through focus groups and written questionnaires at different time points in the HCD process. Three types of data were collected:

Brief feedback questionnaires: All team members (community and support team members) completed anonymous, online questionnaires after the ideation phase in order to monitor experiences and gather feedback to guide the remainder of the project (assessment in Appendix B).

Focus group discussions (FGDs): Sixteen CBD team members participated in FGDs during the implementation phase of the project. The aim of the FGDs was to explore their experiences during the HCD project (see Appendix C for the FGD guide). Separate FGDs were conducted with the adolescent team members (one FGD) and parent team members (two FGDs to enable parents to participate in either French or English). FGDs lasted approximately one hour and were conducted via Zoom by an experienced qualitative researcher not otherwise involved with the project. Audio recordings were transcribed by an external consultant.

Written open-ended questionnaires: The principal support team members (4 CBD team mentors, 1 project coordinator, 1 researcher, and 2 HCD specialists) completed an anonymous written questionnaire (see Appendix D) at the end of the project. The aim was to explore the perspectives of the support team members on strengths and challenges of the ECHO project and the HCD process.

Written questionnaires and FGD transcripts were coded using NVivo 12 qualitative analysis software. Descriptive thematic analysis was used to explore the perspectives of the CBD team members, with distinct exploration of community and support team feedback [28]. To ensure rigor, three members of the research team (K.A, G.F., and B.M.) independently and inductively coded all transcripts. Initial codes were compared and refined during debrief meetings, and themes were developed collaboratively. Triangulation of data sources, including feedback questionnaires, focus group discussions, and written surveys, further enhanced the credibility of the findings.

## 3. Results

### 3.1. Findings from HCD Process

The findings from the HCD process for the four CBD teams are summarized in Figure 1 (for the adolescent-led projects) and Figure 2 (for the parent-led projects). The figures were adapted from the ECHO project’s final report that was prepared for community partners, including participating schools, community organizations, and research participants. Each team aimed to develop and implement an intervention to address specific challenges related to vaccine hesitancy within their target population. The adolescent teams focused on combating misinformation and improving health literacy among youth, while the parent teams centered on building trust and addressing concerns specific to parents of young children.
Figure 1Adolescent-led HCD projects: (**a**) Adolescent team 1 (Montréal-Nord): “My Choice for My Community: a series of YouTube videos on COVID-19 prepared by and for youth”; (**b**) Adolescent team 2 (Parc-Extension): “Using educational gaming to combat misinformation: supporting adolescents to make informed decisions during a pandemic”.
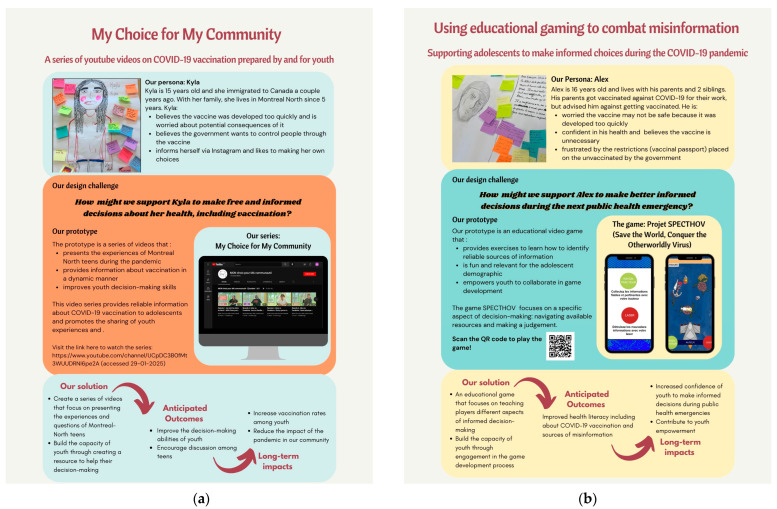

Figure 2Parent-led HCD projects: (**a**) Parent team 1 (Parc-Extension): “Fostering community trust to increase vaccine confidence: an empowering vision for the mothers of Parc Extension”; (**b**) Parent team 2 (Montréal-Nord): “Bringing parents’ voices into child health initiatives: plan to create a parent council for children’s ”.
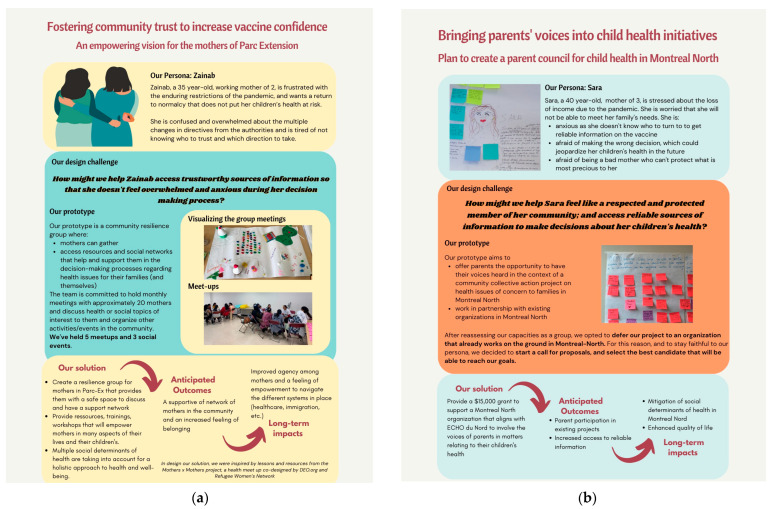


The sections that follow detail the interventions developed by each team, highlighting their design process and key activities.

Adolescent team 1: My choice for my community: a series of YouTube videos on COVID-19 vaccination prepared by and for youth

The adolescent team from Montréal-Nord conducted 11 interviews with unvaccinated youth living in their neighbourhood. Based on the synthesis of these interviews, their design challenge was to support youth in making free and informed decisions about their health, including with respect to vaccination. The team prototyped, tested, and then developed a series of YouTube videos to provide reliable information about COVID-19 vaccination to youth and promote the sharing of youth experiences “https://www.youtube.com/@monchoixpourmacommunaute5668 (accessed on 28 January 2025)”. Additional findings are described in Figure 1a. The team’s implementation budget was used to rent video equipment and hire two university film students to shoot and edit the videos and organize a red-carpet community launch event. The video series and associated discussion guide were shared with schools and research and community partners, as well as via social media, and in two scientific conference presentations.

Adolescent team 2: Using educational gaming to combat misinformation: supporting adolescents in making informed decisions during a pandemic

The team of adolescents from Parc-Extension conducted 13 interviews with unvaccinated youth. They identified misinformation surrounding health issues, including vaccination, as a significant challenge affecting young people. To address this design challenge, they prototyped and then developed an educational videogame intended to help youth navigate different sources of information and make informed decisions. A local start-up specializing in educational game development was contracted to create the game, with implementation funds used to cover development costs and to host a community launch event to showcase the project. Figure 1b provides additional information. Following the initial development of the game, the team conducted interviews with users, receiving many positive comments and useful feedback for improving various elements of the game.

Parent team 1: Fostering community trust to increase vaccine confidence: an empowering vision for the mothers of Parc-Extension

This team of mothers from Parc-Extension conducted 11 interviews with parents living in their neighbourhood who had unvaccinated children between the ages of 5 and 11. Synthesis of the interviews highlighted the anxiety many parents felt about having their children vaccinated against COVID-19, as well as the significant economic and social challenges faced by parents, especially immigrants and newcomers, during the pandemic. The team designed a community resilience group to support mothers in accessing resources and social networks to aid in the decision-making process (additional findings are shown in Figure 2a). The project launched with a celebration on Mother’s Day, and the group continued to hold monthly meet-ups and cultural events for six months, though not beyond as the mothers had hoped. Implementation funds were used to host these meetings and events, covering costs for space and equipment rental as well as refreshments. The team’s work was featured in a news article published by the Canadian Broadcast Corporation [29].

Parent team 2: ECHO du Nord parents: Bringing parents’ voices into child health initiatives in Montréal-Nord

In Montréal-Nord, the parent team conducted six interviews with parents who had unvaccinated children. Recruitment of interview participants was more challenging for this team, as they frequently encountered a general distrust of initiatives linked to government or outsiders (e.g., university researchers). The team identified a lack of community belonging and trust as barriers to making informed decisions about vaccination for children and youth. They designed an ambitious proposal to initiate a parent’s council for the neighbourhood, intended to provide an opportunity for parents to have their voices heard in the context of a community collective-action project. The proposed intervention aimed to represent parents’ voices in dialogue with local health authorities and decision-makers on issues related to the health of their children and adolescents. The team was inspired by the ladder of citizen participation [30], and they aimed to transform public participation from being performative to a genuinely participatory model, helping parents feel a true sense of belonging and ownership in decisions influencing their children’s health.

The four CBD team members planned to remain involved until the council was established and running, with the participation of more community members. However, the team decided to change course after realizing the time commitment required and opted to outsource the implementation of the initiative by providing a grant to a local community-based organization (see Appendix B for description). Unfortunately, no applications were received for the grant, and the council was not launched. Only a small portion of the implementation funds was used for one community meeting. The remaining funds were instead used to support project-associated conference participation and publication fees.

### 3.2. Process Evaluation Results

The process evaluation was iterative, designed to learn and adapt as the project progressed. The aim was to answer two primary questions: (1) How feasible is the HCD approach for addressing vaccine hesitancy through community-driven interventions, and (2) what factors, including team dynamics and disciplinary tensions, influenced the success of the HCD process? FGDs and surveys were conducted at strategic intervals to capture team perspectives at key points in the process, such as the conclusion of each HCD phase. This iterative approach allowed for timely feedback to inform subsequent phases while also enabling an overall evaluation of the project’s implementation and process outcomes.

The process evaluation results of the ECHO project revealed three central themes that emerged from the questionnaires’ answers and FGDs with CBD teams.

Theme 1: Appreciation of the project’s diversity, respectful team dynamics, and collaborative learning approach

Among the most appreciated aspects of the ECHO project was the diversity of its teams in terms of age, gender, ethnocultural identity, and educational background. As one parent stated, “What I really liked, and I had told my colleagues at the beginning, is the diversity in our team. Whether it’s in terms of our professional backgrounds, or our approaches as well. […] Culturally we are also different. We also have gender parity. I loved that. In my life, in general, I don’t always have this diversity around me, so I really liked that”.

Despite the diversity of backgrounds and opinions, the CBD team members reported feeling respected and comfortable sharing their points of view during the HCD process. According to one parent, “We didn’t mind telling each other things in meetings, how we saw things, how we wanted things to be oriented. So that was good. What I appreciated about our mentors was that they were democratic, even if what we said didn’t always go their way, they gave us the opportunity to express our opinions”. The support of mentors was especially important for adolescent researchers, building their confidence and helping them overcome initial apprehensions about participating in the project. Speaking about their mentor, one adolescent said, “She taught us a lot and she helped us in everything we did. Sometimes we had little moments of inattention, little delays, and she knew how to put us on the right path for the future”.

Many of the CBD team members expressed a strong sense of pride over and ownership of the work accomplished and the interventions developed. Adolescent researchers, in particular, appreciated learning various research skills that enabled them to explore the perspectives of their peers. One adolescent made the following remark: “…the thing I liked the most was interviewing teenagers. When we did little interviews on Zoom, it helped me a lot to be less self-conscious, to be open-minded, to understand people of my age”.

Theme 2: Support for core HCD values and approach

Many team members highlighted the values of creativity, collaboration, and empathy as being unique strengths of the HCD process. One adolescent endorsed the process of HCD, stating “…because it is a way to study the needs and fears of the community and then respond to them by concentrating the information and making something innovative,” while a support team member reflected on the advantages of “…in-person teamwork sessions during which we could feel the enthusiasm of the team members and see the fruit of their brainstorming”.

The development of personas to support the synthesis of interview data was described as particularly helpful in humanizing reflections during the search for solutions. A support team member identified HCD as “…a new approach for me, and it was very interesting to learn about. I liked that the approach is centered around a person rather than data which allows for a holistic approach to implementing solutions”. CBD team members also found the persona approach useful, with one parent reflecting that “…when we were thinking about our final solution and we were a little bit lost, we went back to the persona, and it basically summed up what we had seen over the course of two months and there is something really strong in there.”.

However, synthesizing interview findings into a persona was also identified as a challenging task. One adolescent researcher reported that “…one of the difficulties is the fact that each person was different, and we had to take the information about everyone and create a persona. So, since everyone was so different and so varied, it was hard to manage that”. A support team member suggested that more interviews may have allowed for distinct identification of other personas: “The fact that there are multiple personas in a target community, but we only had one for each, makes one wonder about the scope of the solutions in the neighborhoods”.

Theme 3: Disciplinary tensions between HCD and community health research

The feedback from the design thinking (*n* = 2) and public health (n = 6) specialists highlighted some tensions between the disciplinary approaches. Public health researchers and graduate students tended to describe HCD as overly “business-oriented” and lacking some of the rigor they are used to in community-based and qualitative research. Many suggested that greater integration of participatory and qualitative research approaches would have strengthened the HCD process. One public health support team member made the following comment: “I had hoped we could learn the HCD approach and adapt it to this community participatory research project by integrating useful HCD elements alongside other approaches—for example, community needs assessment, more structured qualitative analysis of interview data, building genuine research-community partnerships”. However, HCD specialists felt their methodology needed to be more fully embraced, noting that “the mindset in the support team was ‘We’ll accept the HCD tools that make fit our paradigm and discount the ones that don’t.’”.

HCD experts were also concerned about a perceived “lack of trust in the process and a low tolerance for ambiguity that are pretty critical components of having successful outcomes”. Indeed, the CBD team members and mentors reported finding it challenging at times to embrace uncertainty in the HCD process. While the entire HCD process was presented briefly at the outset of the project, the specific goals of each step and their implications for subsequent phases were detailed progressively throughout each training session, which preceded the next HCD phase. One parent remarked that “…it was as if at each step we opened a little bit the Pandora’s box so that we could look inside. But it doesn’t work like that. […] I think that from the beginning we have to clarify things. We have to know the steps and then we know how to align ourselves and how to do it. But we were a bit like blind men on the way and at one point we had the impression that the solutions did not come from us …”.

## 4. Discussion

The ECHO project provides valuable insights into addressing vaccine hesitancy through the implementation of HCD within a community-participatory research framework. Our findings indicate that HCD can be used to develop tailored interventions that incorporate the experiences and preferences of diverse community members, potentially fostering trust and cultural relevance. This supports existing research on participatory approaches, which emphasize the benefits of community engagement for building trust and ensuring interventions are culturally appropriate [31]. However, this use of HCD in applied public health research has also revealed some limitations and challenges.

One of the key strengths of the ECHO project was the fact that distinct approaches were taken by the parent- and youth-led teams, highlighting the flexibility and adaptability of HCD. The parent-led teams focused on building trust and providing support networks for parents who were hesitant about vaccinating their children. For instance, the community resilience group in Parc-Extension aimed to address parental anxieties by empowering them through information sharing and social support. This aligns with existing research indicating that parental vaccine hesitancy often stems from concerns about safety, efficacy, and potential impacts on their children [32,33].

Conversely, the youth-led teams designed interventions to combat misinformation and provide accurate information among their peers. Adolescents, who typically rely on distinct information channels and seek advice from peers and social media, were targeted with initiatives like the YouTube video series and educational gaming. These interventions were intended to improve health literacy and decision-making skills, directly addressing the misinformation that fuels vaccine hesitancy within this demographic. Research has shown that accurate and relatable information is crucial in influencing adolescent health behaviors [34,35].

Throughout the ECHO project, the core values of HCD—creativity, collaboration, and empathy—were widely shared by team members. The process of creating personas to synthesize interview data was particularly beneficial in humanizing the design process and guiding the development of solutions. Personas helped the team conceptualize and address the specific needs and behaviors of community members, ensuring that interventions were grounded in real-world contexts. However, research has also highlighted challenges in creating personas within HCD, particularly in ensuring inclusivity and accurately representing diverse user needs, including in public health contexts [36]. To streamline the design process within the limited project timeframe, the teams were advised to develop a single persona. While this approach allowed for a focused intervention, it also restricted the representation of the diverse perspectives within the community. Future projects could benefit from allocating additional time during the inspiration phase to conduct more interviews and create multiple personas, ensuring a broader range of community voices are captured.

Existing research finds that integrating HCD into traditional public health research can present some challenges due to the differing objectives and methodologies of each approach [10,12]. HCD focuses on rapidly prototyping and iterating solutions to specific problems within a shorter timeframe, prioritizing immediate and actionable outcomes. In contrast, community-based participatory research (CBPR) emphasizes generating local, contextual insights over longer period, with the goal of building community capacity and informing future actions. Balancing HCD’s rapid, solution-oriented nature with CBPR’s longer-term, partnership-building approach can create practical tension, particularly in aligning priorities and timelines.

The limitations of HCD were particularly noticeable in the Montréal-Nord parents’ initiative, wherein the team developed a customized solution but faced challenges in sustaining the project due to the short timeline and the difficulty of maintaining long-term engagement from team members. Similarly, the Parc-Extension mothers’ project struggled to maintain momentum because of unresolved tensions with existing community organizations. To address these challenges in future projects, projects could explore extending timelines or establishing sustained partnerships with local organizations to ensure continuity. Moreover, the diversity of team members, noted as a strength of the project, could be further leveraged by integrating additional equity-focused training or mentorship opportunities to deepen collaboration and enhance outcomes. Additionally, public health researchers expressed concerns about the perceived “business-oriented” nature of HCD and its differences from rigorous qualitative and CBPR methods. Conversely, HCD specialists emphasized the need for a more complete embrace of the methodology, highlighting the importance of trust in the process and tolerance for ambiguity. These tensions reflect broader challenges in integrating HCD into established public health research frameworks, as noted in other studies [10,12]. Future research could explore interdisciplinary dialogue to bridge these differences and investigate hybrid approaches that leverage HCD’s creativity and rapid prototyping alongside CBPR’s rigor and community-driven focus. While challenges remain, emerging examples of integrated approaches suggest that combining the strengths of both methodologies is feasible and holds promise for addressing complex public health issues [37].

This study was subject to several limitations that could impact the interpretation and applicability of its findings. First, there was no formal measure of the effectiveness of the interventions, making it challenging to assess their impact on vaccine confidence within the communities. Second, the study’s timeframe was a constraint, potentially limiting the development and full implementation of the interventions. Third, public health restrictions significantly reduced in-person engagement during the HCD process, which is crucial for building trust and understanding with community participants. Finally, the evolving nature of the science surrounding COVID-19 vaccination during the study period may have influenced the participants’ perceptions about the relevance of getting vaccinated and thus of the interventions themselves, thereby complicating both the design and evaluation phases of the project.

## 5. Conclusions

The ECHO project highlights the promise and challenges of integrating HCD into public health initiatives. By addressing the unique needs of different communities and fostering inclusive, collaborative environments, HCD can contribute to the design of more effective and equitable health interventions. However, careful consideration of the practicalities of scaling and adapting these interventions is crucial. Continued exploration and documentation of integrated HCD and CBPR approaches will be essential for advancing our understanding of how to best utilize these methodologies to improve public health outcomes and address vaccine hesitancy on a broader scale.

## Figures and Tables

**Table 1 ijerph-22-00198-t001:** Summary of objectives, key activities, and training topics during the inspiration, ideation, and implementation phases of the human-centered design process.

**Objectives**	**Key Activities**	**Training Workshop Topics**
Phase 1: Inspiration		
To understand and build empathic narratives about COVID-19 vaccine hesitancy To generate a wide array of solutions that portray the perspectives of the the user and meet their needs	Hold interactive virtual sessions to learn about vaccine hesitancy from substantive experts (with backgrounds in social sciences, clinical medicine, and public health). Conduct semi-structured interviews with unvaccinated youth and parents of unvaccinated children. Facilitate brainstorming sessions to synthesize interview data and create a persona that captures the needs and motivations of a group of users with shared characteristics. Frame the design challenge using “how might we” questions (e.g., how might we help our persona access trustworthy sources of information about COVID-19 vaccination)	Training 1: November 2021 Introductions, description of the study objectives, participatory research principles, ethics. Training 2: December 2021 Creating interview guides and conducting interviews Training 3: January 2022 Synthesizing findings from interviews Training 4: February 2022 Creating personas and framing a design challenge
Phase 2: Ideation		
To create an initial prototype of the solution and gather input from users To use an iterative process to refine the prototype based on user feedback	Brainstorm solutions to the design challenge and vote to prioritize potential solutions for prototyping stage Create an initial prototype using a storyboard or model Test prototypes, capture feedback, and make modifications based on user suggestions	Training 5: March 2022 Prototyping initial solutions
Phase 3: Implementation		
To create and test a solution in real-world conditions	Plan for implementation pilot using a social business model canvas (in regard to, e.g., key resources, stakeholders, activities, cost, etc.) Create an evaluation plan to assess strengths, limitations, and key outcomes Implement and gather feedback from participants	Training 6: April 2022 Planning for piloting, social business plans, monitoring, and evaluation

## Data Availability

Data are unavailable due to privacy and ethical restrictions.

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
