# Peer review of "Using Human-Centered Design in Community-Based Public Health Research: Insights from the ECHO Study on COVID-19 Vaccine Hesitancy in Montreal, Canada"

_ijerph, 2025, doi:10.3390/ijerph22020198_

Round 1

Reviewer 1 Report

Comments and Suggestions for Authors

Reviewer 2 Report

Comments and Suggestions for Authors

Background:

 - Page 2, Para 2: Perhaps also integrate literature re: the role of mis-information and dis-information + VH?

 - Para 3: Consider adding "results reported elsewhere" for Obj 1, and include citation? 

Methods

 - Lines 88-89 - add reference?

 - Lines 95+96 - add reference?

 - Line 107 - add how many people supported each team - looks like it was 5+. Is it possible to clarify what you called this group (e.g., support staff?) so that when we read "the teams" in the next paragraphs, we clearly know if that's the 4-person community team, the 8-person community group, or if that also includes the support staff? Also, is it worth adding a note re: how the adolescent and parent teams were related/not and did/not work together?

 - Line 119 - citation out of [  ]. Also, leading up to this, even thought the methods are described elsewhere, and likely the tools referenced, I would be in support of some more details and links to the tools/guides. Also, can you add detail re: how long each of these phases took?

 - Line 136 - can you take this sentence up one level? To me, documentation and critical reflection are processes you used to learn/achieve/understand something bigger - can you list that?

Results

 - Line 164+ - could you add a 2-3 sentence summary here re: what will be presented next? I spent time reviewing the figures, and was then confused by the narrative that followed. Maybe just re-introduce what the goals were (each team to develop an intervention) and then introduce that you will be presenting X and then Y? I suggest putting the figures after line 200.

 - 3.2 Process Evaluation - in reading this section it strikes me that you likely had some a priori ideas re: processes and how to support success. In the Background and Methods, might it be valuable to state the questions you were hoping to be able to answer, and why FGDs and surveys at that frequency were chosen?

 - Line 272 - just double check. If the concept and process and reasoning of 'personas' is not mentioned in the background or methods section, consider adding it to provide context for this.

 - Lines 289+ - is it worth adding how many of your team fall into each of those camps? 

 - Theme 3 as a whole, and Lines 312-313 -just double check that you use this as a "note" to make an inference/recommendation about? This is awesome!

 - Line 345 - add in a sentence/link to literature here?

 - Line 352 - Strong comment. Add literature in here? Or pose as a question/area of exploration? unclear if this is a claim based on your work or lit base.

** I suggest adding in some recommendations in at the end of the Discussion. 

Round 2

Reviewer 2 Report

Comments and Suggestions for Authors

Great edits!